# Thermally Coupled NTC Chip Thermistors: Their Properties and Applications

**DOI:** 10.3390/s24113547

**Published:** 2024-05-31

**Authors:** Milan Z. Bodić, Stanko O. Aleksić, Vladimir M. Rajs, Mirjana S. Damnjanović, Milica G. Kisić

**Affiliations:** 1Faculty of Technical Sciences, University of Novi Sad, 21102 Novi Sad, Serbiamirad@uns.ac.rs (M.S.D.);; 2Institute of Nuclear Sciences—INN Vinca, 11351 Belgrade, Serbia

**Keywords:** chip thermistors, thermal junction, heat transfer

## Abstract

Negative temperature coefficient (NTC) chip thermistors were thermally coupled to form a novel device (TCCT) aimed for application in microelectronics. It consists of two NTC chip thermistors Th_1_ and Th_2_, which are small in size (0603) and power (1/10 W). They are in thermal junction, but concurrently they are electrically isolated. The first thermistor Th_1_ generates heat as a self-heating component at a constant supply voltage U (input thermistor), while the second thermistor Th_2_ receives heat as a passive component (output thermistor). The temperature dependence R(T) of NTC chip thermistors was measured in the climatic test chamber, and the exponential factor B_10/30_ of thermistor resistance was determined. After that, a self–heating current I_1_ of the input thermistor was measured vs. supply voltage U and ambient temperature T_a_ as a parameter. Input resistance R_1_ was determined as a ratio of U and I_1_ while output thermistor resistance R_2_ was measured by a multimeter concurrently with the current I_1_. Temperatures T_1_ and T_2_ of both thermistors were determined using the Steinhart–Hart equation. Heat transfer, thermal response, stability, and inaccuracy were analyzed. The application of thermally coupled NTC chip thermistors is expected in microelectronics for the input to output electrical decoupling/thermal coupling of slow changeable signals.

## 1. Introduction

A thermistor is a type of resistor, the resistance of which can have a positive temperature coefficient (PTC) or negative temperature coefficient (NTC), i.e., it increases or decreases its resistance with the temperature increase. Typical thermistor geometries are the disk, chip, and thick film. The resistance values, temperature range, and temperature coefficient B depend on the semiconductor oxides used for their preparation [1,2]. They are mass-produced by pressing thermistor powders in molds and after that, sintering compacts the thermistor at different temperatures in air [3,4]. NTC thermistors are divided into two groups: low temperature application range thermistors (−50 °C to +250 °C) [5,6] and high temperature application range thermistors for +250 °C to +1000 °C [7,8]. The semiconductive oxides used in the production of low temperature application range NTC thermistors are based on nickel manganese spinel and its modifications such as NiMn_2_O_4_, (NiMn)_3_O_4_, (NiMnCo)_3_O_4_, (NiMnFeCo)_3_O_4_, and (Fe,Ti)_2_O_3_. They are usually doped by a low percentage of oxides such as CuO, ZnO, CoO, LiO, RuO_2,_ ZrO_2_, Y_2_O_3_, and other oxides to form complex thermistor materials [9,10,11,12,13,14]. The nickel manganese spinel is partly inverse as the tetrahedral ion Ni^2+^ and octahedral ion Mn^3+^ can exchange their sites in the lattice. Octahedral ions can change their valence such as 2Mn^3+^→(Mn^2+^, Mn^4+^). In that way, the Mn^2+^ ion fits the vacancy in the lattice where ion Ni^2+^ is missing. The main carriers in NiMn_2_O_4_ are small polarons (partly polarized) electrons and the energy gap has a relatively low value (around 0.5 eV) [15,16,17,18]. The disk and chip NTC thermistors for the low temperature application range are pressed out of NiMn_2_O_4_ powder in molds and sintered at 1100 °C to 1200 °C/2 h in air [19,20]. The electrodes for disk and chip thermistors are printed out of thick film conductive PdAg paste on both sides. After that, they are dried at 150 °C/10 min and sintered at 850 °C/10 min in air [21,22]. The semiconductive oxides used in the production of NTC thermistors fora higher temperature application range are based on BaTiO_3_, Fe_2_TiO_5,_ Co_3_O_4_, CaTiO_3_, Mg(Al_1−x_Cr_x_)_2_O_4_, Bi_2_Zr_3_O_7_, ZrO_2_/CaO, and complex oxides such as MgAl_2_O_4_–LaCr_0.5_Mn_0.5_O_3_, Ni_1.0_Mn_2−x_Zr_x_O_4_, Ce_1.2(1−x)_Mn_1−x_Si_x_, Sr_7_Mn_4_O_15_, etc. [23,24,25,26,27,28,29].

A chip thermistor is a leadless component designed for surface mounting technology (SMT). It is small in size and small in power and serves for the measurement of the temperature in the measuring point in air, liquids, the ground, and solid bodies. Their outlook and thermal coupling are given in Figure 1.

A novel chip TCCT device is based on NTC chip thermistors. The thermistors were joined laterally using epoxy resin, i.e., they are thermally coupled through epoxy but they are electrically decoupled (galvanically insulated). The input thermistor Th_1_ serves as a self-heating thermistor powered by DC or AC voltage U, while the output thermistor Th_2_ acts as a heat receiver and changes its higher electrical resistance R_2_. The main property of the chip TCCT device is that the temperature of the output chip thermistor follows the temperature of the input self-heating chip thermistor. Moreover, the input and output thermistors are galvanically isolated and the electrical noise caused by switching power electronics is not transferred from the input to output thermistor. The comparatively low (electrical) coupling capacity between the two NTCs and (2) their thermal inertia allows (1) to reduce the disturbing influence of the voltage or current peaks on the temperature measurement and (2) to smoothen the ripple of the temperature signal. The output thermistor can be connected to an operational amplifier and used in different ways: feedback to input power, output temperature to input power convertor, electro-thermal potentiometers, etc. In comparison to the thick film (TF) TCT device with segmented thermistors and disk TCDT device which were realized recently in our previous work [30,31], the TCCT device has the following advantages: it occupies the smallest surface and has a simpler construction, lower cost, smaller power, and a high availability of chip thermistors with different nominal resistances, which is suitable for the realization of different resistance ratios in a thermal coupling device.

## 2. Experiment and Results

### 2.1. Main Properties of Chip Thermistors

The chip thermistors given in Figure 1a (EPCOS NTC 0603, TDK Europe, Munchen, Germany) have the following main properties: a dissipation power of 1/10 W, nominal resistance of around 2600 Ω at a room temperature of 25 °C, and exponential factor B_25/75_ ≈ 3900 K (given by the producer). For our experiments, nominal resistance was additionally measured at 20 °C and B_10/30_ was determined in the range of 10 °C to 30 °C. Electrical resistance R was measured in the climatic test chamber with the temperature increase T as the temperature behavior R(T) of the NTC chip thermistor and given in Figure 2. It can be approximated bythe Steinhart–Hart equation in the first article (1) [32]:
(1)$$R(T)=R_{0}\exp\left[-B\cdot (\frac{1}{T_{0}}-\frac{1}{T})\right]$$

Resistance R_0_ is the nominal value of the chip resistor at 20 °C, T_0_ is the room temperature at 20 °C (293,16 K), B is the exponential temperature coefficient of the chip thermistor, and T is the current temperature of the thermistor. The thermistor exponential factor B was obtained in the vicinity of room temperature using the values of chip resistance R_10_ and R_30_ measured at temperatures T_10_ = 10 °C and T_30_ = 30 °C, respectively. The thermistor exponential factor B is derived from Equation (1) and given in Equation (2):(2)$$B=\left[\frac{T_{10}\cdot T_{30}}{T_{30}-T_{10}}\right]\cdot \ln\left[\frac{R_{10}}{R_{30}}\right]$$

Using Equation (2) and the results in Figure 2, the chip thermistor coefficient was determined as B_10/30_ = 3908.3 K. Nominal resistance at 20 °C was measured with the digital multimeter Fluke 179 as R = 2680 Ω. Also, using Equations (1) and (2), the unknown current temperature T of the thermistor is a function of current thermistor resistance R. In that way, the temperature T_1_ of the self-heating thermistor Th_1_ (input thermistor) is a function of resistance R_1_, coefficient B_1_, and can be calculated by Equation (3):(3)$$T_{1}=\left[\left(B_{1}\cdot T_{0}/(B_{1}+T_{0}\ln(\frac{R_{1}}{R_{01}}))\right)\right]$$
whereas resistance R_1_ = U/I_1_ is the resistance of the self-heating thermistor Th_1_ (input resistance) and R_01_ is the nominal value of thermistor Th_1_ measured at 20 °C. Hence, thermistor Th_2_ in the thermally coupled device has temperature T_2_ calculated by thermistor resistance R_2_, coefficient B_2_, and is given in Equation (4):(4)$$T_{2}=\left[\left(B_{2}\cdot T_{0}/(B_{2}+T_{0}\ln(\frac{R_{2}}{R_{02}}))\right)\right]$$

The electrical resistance R_2_ of thermistor Th_2_ is measured using a multimeter in the climatic test chamber, and R_02_ is the thermistor nominal value measured at 20 °C. In our partial case, Th_1_ and Th_2_ were made out of the same material and B_1_ = B_2_ = B and R_01_ ≈ R_02_. Before the measurement of thermistor resistance, the climatic test chamber was twice recalibrated by a PT-1000 platinum thermometer to lower the inaccuracy of measuring temperature T to around ±0.020 °C.

### 2.2. Thermal Coupling of Chip Thermistors

At first, thin wire leads were soldered onto the chip thermistors and then the chip thermistors were insulated using epoxy resin. Finally, a pair of insulated chip thermistors were placed in lateral contact and joined together using a thin layer of epoxy between them to form the TCCT device as given in Figure 1b,c. The main properties of the TCCT device were measured in the climatic test chamber. The measuring setup is given in Figure 3; it consists of thermally coupled chip thermistors TCCT and three multimeters used for the measuring of the input supply voltage U, self-heating current I_1_ of thermistor Th_1_, and output resistance R_2_ of thermistor Th_2_.

The self-heating current I_1_ of thermistor Th_1_ in the TCCT device vs. supply voltage U was measured at different ambient temperatures T_a_ as a parameter in the range from 0 °C to 40 °C (in steps of 5 °C). The supply voltage U was increased in steps of 1 V each 20 s to observe the behavior of the self-heating process. The input power P_1_ = U∙I_1_ of the self-heating thermistor Th_1_ was defined for different ambient temperatures T_a_ in the climatic test chamber, which was also changed in steps of 5 °C. The results of these measurements are given in Figure 4.

### 2.3. Resistances and Temperatures in TCCT Device

The resistance R_1_ of the chip thermistor Th_1_ is defined from the self-heating current I_1_ and supply voltage U as R_1_ = U/I_1_, while the resistance R_2_ of the chip thermistor Th_2_ is measured by a multimeter at the same time as U and I_1_. The results of these measurements are given in Figure 5.

The temperatures T_1_ and T_2_ of the chip thermistors Th_1_ and Th_2_, respectively, are defined using Equations (3) and (4), and the values of the resistances R_1_ and R_2_ are given above in Figure 5. The ambient temperature T_a_ was used a parameter. The obtained results are given in Figure 6.

### 2.4. Self-Heating Current Stability vs. Time of TCCT Device

Using the same measuring setup as given in Figure 3, the behavior of the self-heating current I_1_ of the chip thermistor Th_1_ vs. time t from the switch on moment (t = 0) of the fixed supply voltage U was measured. The supply voltage U was changed as a parameter in steps of 1 V in the climatic test chamber. The I_1_ curves from the switch on to saturation level are given in Figure 7. The delay time from the switch on supply voltage U marked as t_d_ ≈ 30 s was estimated from the behavior of the self-heating current I_1_(t) in Figure 7 in the region below the knee for a bundle of curves. The criterion of the stability of I_1_ in the horizontal region was introduced as k = I(t = 40)/I(t = 80) > 0.97. Using this criterion of stability, the bar diagram in Figure 7 was formed for the values of the supply voltage U and self-heating current I_1_ to follow the criterion k > 0.97 in the ambient temperature range from 0 °C to 40 °C.

The self-heating cycle from t = 0 s to t = 180 s and the cooling cycle from t = 180 s to t = 360 s for the TCCT device was measured using the same measuring setup as given in Figure 3 and the fixed input supply voltage U = 8 V at ambient temperature T_a_ = 20 °C. The resistances R_1_ and R_2_ and temperatures T_1_ and T_2_ of the chip thermistors Th_1_ and Th_2_, respectively, are given in Figure 8.

### 2.5. Application of TCCT Device

The applications of the TCCT device with thermally coupled/electrically insulated chip thermistors are related to microelectronics. For example, two applications with TCCT are proposed and given in Figure 9. Block schemes A and B can be classified as voltage dividers/slider-less potentiometers or an electro-thermal potentiometer limited with a factor of stability of k > 0.97 (without a moving part known as the slider) based on chip thermistors Th_1_ and Th_2_, and the input voltage U on thermistor Th_1_ governs with electrical resistance R_2_ in thermistor Th_2_ via generated and transferred heat (thermal route). In the other block scheme in Figure 9, the resistance R_2_ of thermistor Th_2_ of the TCCT device is placed in a modified bridge with an operation amplifierIC_1_ (TLV 9002) for the linearization of the output V_out_. The fixed resistor in the bridge, such as the 1.5 kΩ resistor placed in parallel with a 1 kΩ resistor +5 kΩ potentiometer, is chosen based on the nominal value of the NTC chip thermistor Th_2_ and the temperature range. A parallel connected capacitor such as the 100 nF with a fixed resistor of 1.5 kΩ (input to output feedback) limits bandwidth, improves stability, and helps to reduce noise. The range of temperature T_2_ linearization is room temperature 20 °C ± 25 °C. The bridge enables a temperature to voltage conversion V_out_ = F(T_2_) and power to voltage conversion V_out_ = F(P_1_). Other applications are related to the control of small power AC/DC converters, thermostat function in air conditioning equipment, level detection, LED switching, etc.

The electro-thermal potentiometer functions A and B are given in Equations (5) and (6), respectively:(5)$$A:V_{out}=\frac{R_{0}}{R_{0}+R_{2}}\cdot V$$
(6)$$B:V_{out}=\frac{R_{2}}{R_{0}+R_{2}}\cdot V$$

Output resistance R_2_ = f(U), R_0_—chosen fixed resistor, R_0_ = R_2_ only at room temperature.

The input supply voltage U governs with the self-heating current I_1_ in the thermistor Th_1_ and generates heat P_1_ transferred to the thermistor Th_2_, which governs with electrical resistance R_2_, and output temperature T_2_. Consequently, T_2_ is a function of P_1_ and P_1_(T_2_) given in Figure 10, and is used to measure input power as a function of output temperature. Hence, the TCCT in the bridge configuration enables a temperature to voltage conversion V_out_ = F(T_2_) and power to voltage conversion V_out_ = F(P_1_).

The function P_1_ = F(T_2_) given in Figure 10 consists of measured curves (solid curves), for which each step of ambient temperature Ta was around 5 °C, and interpolated curves, each 1 °C of T_2_ as auxiliary curves (dashed curves). They were interpolated between two measured (solid) curves by using a polynomial of the third order as given in Equations (7) and (8).
(7)$$P_{1}=a_{0}+a_{1}\cdot T_{2}+a_{2}\cdot T_{2}^{2};\;T_{a}=25\;{}^{\circ}C$$
(8)$$P_{1}=b_{0}+b_{1}\cdot T_{2}+b_{2}\cdot T_{2}^{2};\;T_{a}=20\;{}^{\circ}C$$


The dashed curves are introduced gradually with P_n_ polynomials n = 1,2, … 4, as given by Equation (9):(9)$$P_{1n}=(b_{0}+n\cdot \Delta_{0})+(b_{1}+n\cdot \Delta_{1})\cdot T_{2}+(b_{2}+n\cdot \Delta_{2})\cdot T_{2}^{2}$$
$$\Delta_{0}=(a_{0}-b_{0})/5;\;\Delta_{1}=(a_{1}-b_{1})/5;\;\Delta_{2}=(a_{2}-b_{2})/5$$
∆—increments of polynomial coefficients: n = 1,2 … 4.

The function P_1_ = F(T_2_) given in Figure 10 consists of measured curves (solid curves), each around 5 °C of ambient temperature Ta, and interpolated curves, each 1 °C of T_2_ as auxiliary curves (dashed curves). They were interpolated between two measured (solid) curves by using a polynomial of the third order as given in Equations (7) and (8).

The described application is related to the control of small power AC/DC converters using the function P_1_ = F(T_2_) given in Figure 10. Other applications have to be developed using the applications’ electro-thermal potentiometers and bridges for the linearization of the output temperature T_2_ in Figure 9.

## 3. Discussion

### 3.1. Operating Point of TCCT Device

Small NTC chip thermistors 0603 (EPCOS), as given in Figure 1, with a dissipation power of 1/10 W, nominal resistance ofaround 2680 Ω at 20 °C, and exponential factor B = 3908 K, were used in the forming of thermally coupled chip thermistors—the TCCT device—for the first time. The slope of the chip thermistor NTC curve is moderate (Figure 2). The input supply voltage U of the self-heating thermistor Th_1_ (Figure 3) enables the self-heating current I_1_ and generates power P_1_, which are of an exponential/logarithmic type, as given in Figure 4. The chip thermistor Th_2_ is only a heat receiver (passive thermistor). The resistances R_1_ and R_2_ of the chip thermistors Th_1_ and Th_2_ are also of an exponential/logarithmic type dependent on the supply voltage U and ambient temperature Ta as a parameter, as given in Figure 5. The temperatures T_1_ and T_2_ of the chip thermistors Th_1_ and Th_2_, respectively, are also dependent on the supply voltage U and ambient temperature T_a_ (as presented in Figure 6). The curves of T_1_ are more exponential compared to the curves of T_2_ as the heat generated in Th_1_ or power P_1_ with input temperature T_1_ is only partly transferred to the thermistor Th_2_ through the epoxy resinto cause an appearance of output temperature T_2_ on the thermistor Th_2_. As a consequence, the chip thermistor Th_2_ (heat receiver) has a much lower temperature T_2_ compared to T_1_. The insulation thickness of around 1 mm consists of two epoxy layers which enable the transfer of a part of the heat from the self-heating to the heat receiving thermistor. The temperature difference ΔT = T_1_ − T_2_ changes from around 2 °C to around 16 °C in the range of the ambient temperature Ta from 0 °C to 40 °C. The temperature difference ΔT is a function of the input supply voltage U as ΔT(U) = T_1_(U) − T_2_(U), and using Equations (3) and (4), it can be defined as in Equation (10):(10)$$\Delta T=B_{1}\cdot T_{0}/(B_{1}+T_{0}\ln(\frac{U/I_{1}}{R_{01}}))-B_{2}\cdot T_{0}/(B_{2}+T_{0}\ln(\frac{R_{2}}{R_{02}}))$$
where the input resistance R_1_ = U/I_1_ and output resistance R_2_ are measured by a multimeter.

### 3.2. Heat Transfer in TCCT Device

The self-heating current I_1_(t) in Figure 7 is a current in a forced regime depending on the fixed supply voltage U. The heat generation and heat dissipation are in balance on the horizontal part of the bundle of curves I_1_(t). The delay time t_d_ ≈ 30 s from the switch on of the supply voltage U was estimated from the behavior I_1_(t) in the region below the knee of curves. As the horizontal curves I_1_(t) at T_a_ = 20 °C are not ideally horizontal, especially for higher values of U, the criterion of the stability of I_1_(t) in the horizontal region was introduced as k = I_1_(t = 40)/I_1_(t = 80) > 0.97 to limit the voltage U and heat generation. Using the same criterion k > 0.97 of stability for other ambient temperatures T_a_ in the ambient temperature range from 0 °C to 40 °C, the bar diagram in Figure 7 was formed with limited values of the supply voltage U and self-heating current I_1_. The aim was to limit the input power P_1_ in accordance with the criterion of k > 0.97. The behavior of the resistances R_1_ and R_2_ and temperatures T_1_ and T_2_ in the TCCT device vs. time given in Figure 8 show that the self-heating cycle (U = 8 V) and cooling cycle (U = 0 V) differ in shape; the self-heating cycle is a forced regime caused by the supply voltage U, and the cooling cycle is a natural process of heat dissipation. The initial heating and cooling time to the heating/cooling balance is a delay time t_d_ ≈ 30 s from the switch on of the supply voltage U.

The heat balance equation in the local equilibrium is given by (11):(11)$$q_{1}=q_{2}+q_{3}+q_{4}$$

The heat generation in the self-heating thermistor Th_1_ is a product of electrical power U·I_1_ and time Δt, as given by Equation (12):(12)$$q_{1}=U\cdot I_{1}\cdot \Delta t$$

The heat transfer from the Th_1_ to Th_2_ chip thermistor through a thin epoxy layer interface d_i_ ≈ 1 mm, surface value A, and thermal conductivity K_e_ is given by Equation (13):(13)$$q_{2}=-K_{e}\cdot A\cdot [(T_{2}-T_{1})/d_{i}]\cdot \Delta t$$

Hence, the sum of the heat loss q_3_ on the boundary of the thermistor Th_1_/epoxy/air corresponding to T_1_ − T_a_ and q_4_ on the boundary of the thermistor Th_2_/epoxy/air corresponding to T_2_ − T_a_ is given by Equation (14).
(14)$$q_{3}+q_{4}=U\cdot I_{1}\cdot \Delta t+K_{e}\cdot A\cdot [(T_{2}-T_{1})/d]\cdot \Delta t$$

The dissipation surface for q_3_ and q_4_ is A_3_ = A_4_ = A_0_ − A, where A_0_ is the total outlook surface of the chip thermistor, T_1_ and T_2_ are the thermistor temperatures, respectively, the epoxy coating thickness of the chip is d_3_ = d_4_ = 0.5 mm, and the heat dissipation length d_a_in the surrounding air is unknown. Further, using an approximation of the temperatures on the epoxy coatings T_e_ as T_e1_ ≈ T_1_ and Te_2_ ≈ T_2_ for the coated chip thermistors Th_1_ and Th_2_, respectively, q_3_ and q_4_ are given as (15) and (16):(15)$$q_{3}=-K_{a}\cdot A_{3}\cdot [(T_{a}-T_{1})/d_{a}]\cdot \Delta t$$
(16)$$q_{4}=-K_{a}\cdot A_{4}\cdot [(T_{a}-T_{2})/d_{a}]\cdot \Delta t$$
where K_a_ is the thermal conductivity of the air. Now, by replacing q_3_ and q_4_ in Equation (14), the heat balance equation is formed at room temperature T_a_ = 20 °C (17):(17)$$-K_{a}\cdot A_{3}\cdot [(T_{a}-T_{1})/d_{a}]\cdot \Delta t+-K_{a}\cdot A_{4}\cdot [(T_{a}-T_{2})/d_{a}]\cdot \Delta t=U\cdot I_{1}\cdot \Delta t+K_{e}\cdot A\cdot [(T_{2}-T_{1})/d]\cdot \Delta t$$

The unknown heat dissipation length *d_a_* in the surrounding air is given by (18).
(18)$$d_{a}=\{U\cdot I_{1}+K_{e}\cdot A\cdot [(T_{2}-T_{1})/d]\}/\{-K_{a}\cdot A_{3}[2\cdot T_{a}-T_{1}-T_{2}]\}$$

Finally, adding the values for U, I_1_, K_a_, K_e_, T_1_, T_2_, T_a_, d_i_, A, and A_3_, the heat dissipation length d_a_ is derived as d_a_ ≈ 38.5 mm in the air at room temperature. The purpose could be that any other heat source within the vicinity of the TCCT (the “size” of this vicinity is defined by d_a_) could affect the characteristic behavior of the TCCT and, therefore, this d_a_ value must at least be roughly estimated. The TCCT device has negligible thermal contact with the board; practically chip thermistors hang on the wires in the air.

The relative thermal coupling k_1,2_ between chip thermistors Th_1_ and Th_2_ can be defined using (19):(19)$$k_{1,2}=[(T_{2}-T_{a})/(T_{1}-T_{a}]$$

The thermal coupling k_1,2_ between the chip thermistors depends on the U, I_1_, and Ta; for example, at a room temperature of 20 °C, U = 8 V, I_1_ = 5.7 mA, its value is k_1,2_ ≈ 0.25. The obtained k_1,2_ value is smaller compared to the thermal coupling in thick film TF TCT and disk TCDT devices due to the fact that chip thermistors have a five times smaller nominal dissipation power.

### 3.3. Comparison of TCCT with TF TCT and TCDT Devices

The properties of the TCCT device were compared in Table 1 with the thick film (TF) TCT device and disk TCDT properties which were published recently in [30,31].

The TCCT is a cubic device and has around 30 times lower volume compared to the TCDT device and around 100 times compared to the thick film TF TCT. The power of TCCT device is 5 times lower and relative thermal coupling k_1,2_ is 1.5 to 2 times lower compared to TF TCT and TCDT devices. This result is a consequence of the heat transfer from the self-heating to passive chip thermistor through the small interface surface and the device body. The surface on the board of the TCCT is around 30 to 100 times smaller and the heat transfer path is only 1 mm. This result of k_1,2_ is a consequence of high alumina thermal conductivity, which is much higher than the thermal conductivity of the epoxy insulation layer between the Th_1_ and Th_2_ thermistors. The TCCT device sensitivity can be defined as Δ_1_ = ΔT_2_/ΔU (from Figure 6) using limitations for U (in Figure 7) and Δ_2_ = ΔP_1_/ΔT_2_ (from Figure 10). In brief, the reason for this is the output to input feedback and practical use of the TCCT for the control of the input power P_1_ by the output temperature T_2_. The comparison of the TF TCT, disk TCT, and TCCT devices’ sensitivity for the ambient temperature T_a_ in the range of around 1 °C to 40 °C is given in Table 2.

The inaccuracy ΔB of the exponential factor B according to Equation (2) given above in Section 2.1 is dependent on the inaccuracies in the measuring of resistances R and temperature T in the climatic test chamber. The inaccuracy ΔB is now determined by Equation (20):Δ_B_ = 2·ΔR/R + 2·ΔT/T(20)

The inaccuracy ΔB measured in the climatic test chamber does not exceed 0.3%.

The temperature inaccuracy Δ_T_ according to Equation (3) depends on resistance R and ΔB from the previous equation. It is defined as in (4):Δ_T_ = 2·ΔR/R + ΔB/B(21)

The temperature inaccuracy Δ_T_ does not exceed 0.5%.

The inaccuracy in the measurement of the self-heating electrical current I_1_ is dependent on the measuring of the input voltage U, ambient temperature T_a_, and time t as I_1_(U, T_a_, t). The self-heating current has an inaccuracy of Δ_sc_ given by the next Equation (22):Δ_sc_ = ΔI/I + ΔU/U + Δt/t(22)

It does not exceed 0.3%. The inaccuracy of the measuring system Δ_MS_ (only digital instruments in Figure 2) depends on the measuring of U, I, R, and the power source Ups, as given in (23):Δ_MS_ = ΔU/U + ΔI/I + ΔR/R + ΔU/U_PS_(23)

The inaccuracy of the full measuring system Δ_MS_ does not exceed 0.4%. After a year of testing, there was no change in the thermistor nominal resistances R_01_ and R_02_ and the exponential factor B.

## 4. Conclusions

In this work, the electrical and thermal properties of the chip thermistor TCCT device (thermally coupled chip thermistors) were measured and analyzed and after that, compared with the previous thick film (TF) TCT device and disk TCDT device (the main properties are given in Table 1 and Table 2). The main advantages are summarized and grouped as follows.

The main properties: The chip TCCT is a resistive device based on the thermal junction of two small NTC chip thermistors: a chip size of 0603 and dissipation power of 100 mW, nominal resistance R_0_ ≈ 2600 Ω, and exponential factor B = 3908 K. The input voltage U was limited using a practical criterion k > 0.97 (Section 2.2, bar diagrams in Figure 7) to keep the stability of the self-heating current according to the ambient temperature T_a_ in the range from around 1 °C to 40 °C. Practically, the self-heating current is limited to a maximum of 10 mA and power to a maximum of 25 mW. The various chip thermistors’ nominal resistances can be chosen to form the TCCT with a resistances’ ratio R_input_/R_output_ from 1:1 to 1:5 or more. The input supply voltage can be DC, AC, or slow changeable impulses, and the output chip thermistor has a thermal response with a thermally coupling factor k_1,2_ ≈ 0.25. The insulation resistance of the epoxy between the chip thermistors was >10^9^ Ω, and the delay time of the chip thermistor response from the switch on of the supply voltage U to a stable self-heating current I_1_ was less than 30 s.The main advantages: The chip TCCT is a small cubic type of device made out of modified nickel manganese chip thermistors (NTC). The thermal coupling (heat transfer) from the self-heating thermistor to heat receiving thermistor is weaker (k_1,2_ ≈ 0.25) compared to the TF TCT and disk TCDT devices (realized recently in our previous works), but the volume and occupied area on a PCB are much smaller. The production of TCCT is cheaper. The TCCT device operates like an electro-thermal potentiometer (slider-less).The core conclusions are the following: The chip TCCT device has a small volume and simple construction, is cheap to use, and enables different applications, such as the measurement of the input power through the output temperature. It can be coupled with operational amplifiers for output linearization and used in microelectronics and power microelectronics as feedback in small power AC/DC convertors, car automation electronics, home appliances, thermo switches, thermostats, air conditioning, etc.

## Figures and Tables

**Figure 1 sensors-24-03547-f001:**
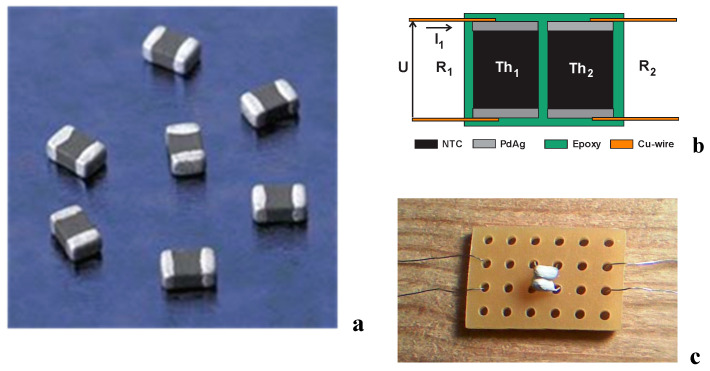
NTC chip thermistors 0603 (EPCOS)—(**a**), block scheme of thermally coupled chip thermistors (TCCT)—(**b**). Th_1_, Th_2_ chip thermistors, R_1_, R_2_ thermistor resistances, U—supply voltage, I_1_—self-heating current—(**c**), top view of encapsulated TCCT placed on a PCB (top view).

**Figure 2 sensors-24-03547-f002:**
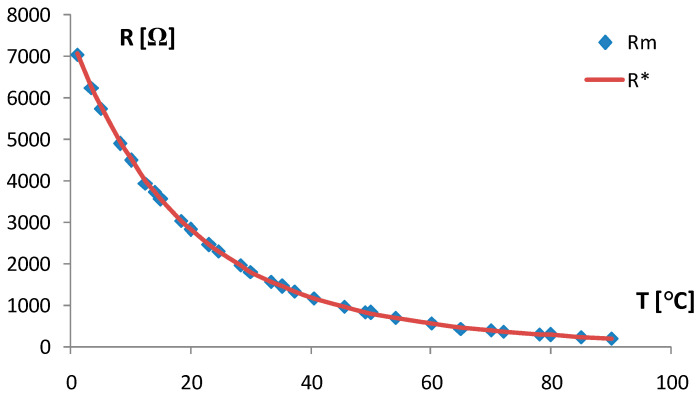
The resistance R of NTC chip thermistor (0603) vs. temperature T measured in the climatic test chamber: R_m_—measured values, R*—fitted curve (solid line).

**Figure 3 sensors-24-03547-f003:**
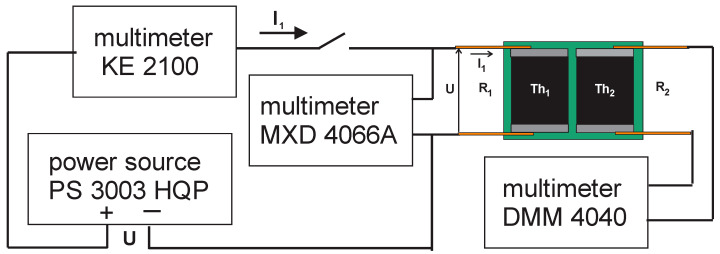
Measuring setup for thermally coupled chip thermistors (TCCT device).

**Figure 4 sensors-24-03547-f004:**
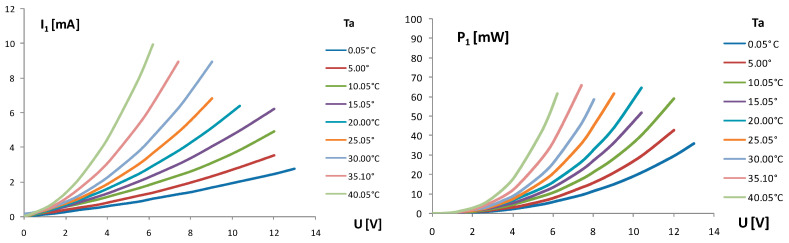
The self-heating current I_1_ and self-heating power P_1_ of the chip thermistor Th_1_ in TCCT device vs. supply voltage U changed in steps of 1 V. T_a_—ambient temperature as a parameter.

**Figure 5 sensors-24-03547-f005:**
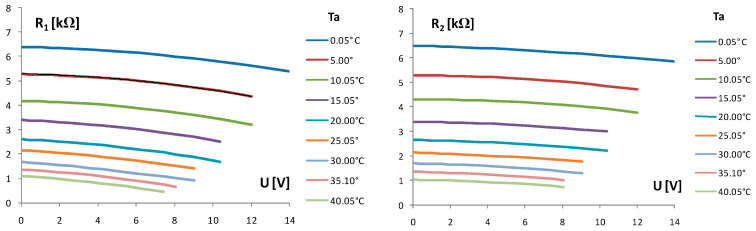
Resistance R_1_ of the self-heating chip thermistor Th_1_ and resistance R_2_ of the chip thermistor Th_2_ in TCCT device vs. supply voltage U changed in steps of 1 V. T_a_—ambient temperature as a parameter.

**Figure 6 sensors-24-03547-f006:**
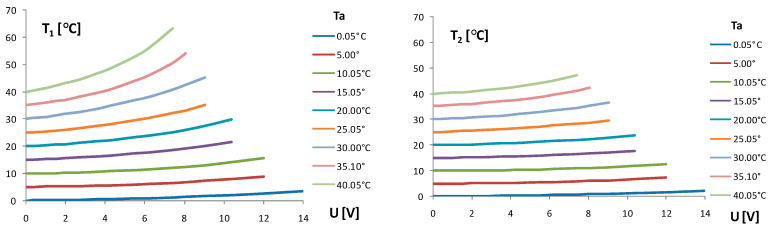
The temperatures T_1_ and T_2_ of the chip thermistors Th_1_ and Th_2_ in TCCT device, respectively, vs. supply voltage U changed in steps of 1 V. T_a_—ambient temperature as a parameter.

**Figure 7 sensors-24-03547-f007:**
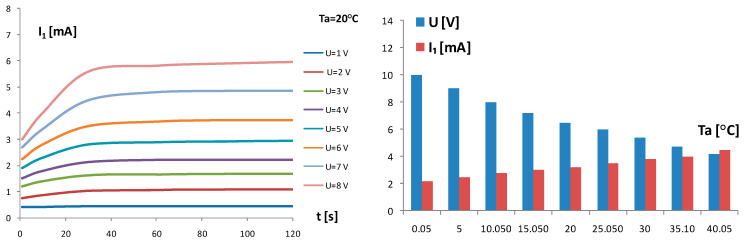
Self-heating current I_1_ of the chip thermistor Th_1_ in TCCT device vs. time t from the switch on of the input supply voltage U. T_a_—fixed ambient temperature. Input voltage U changed in steps of 1 V as a parameter. The bar table for the limitations of U and I_1_ vs. Ta based on the criterion of self-heating current stability k > 0.97 in saturation regime.

**Figure 8 sensors-24-03547-f008:**
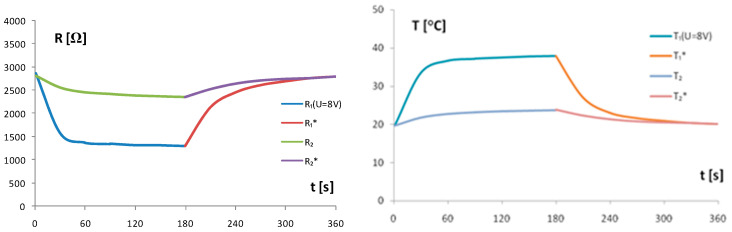
Resistances R_1_ and R_2_ and temperatures T_1_ and T_2_ in the TCCT device for the self-heating cycle from t = 0 s to t = 180 s and cooling cycle (*) from t = 180 s to t = 360 s. Fixed input supply voltage U = 8 V, T_a_—ambient temperature T_a_ = 20 °C. In the self-heating cycle, supply voltage was fixed to U = 8 V, while in the cooling cycle, the supply voltage U is switched off (U = 0).

**Figure 9 sensors-24-03547-f009:**
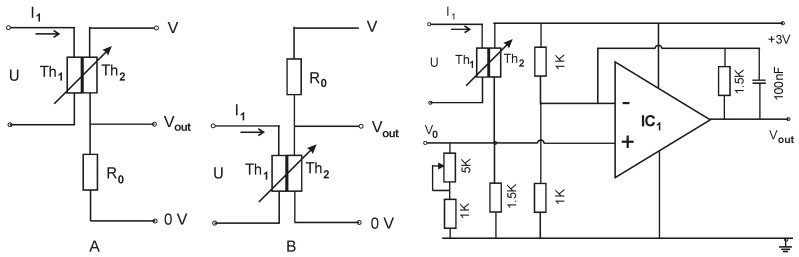
Application of TCCT device: thermo-electrical potentiometers (A and B) with chip thermistors. Th_1_—self-heating thermistor, Th_2_—heat receiver (thermally coupled), U—input voltage, I_1_—input (self-heating) current, V—DC voltage, V_out_—output voltage, R_0_—fixed resistor. Thermally coupled thermistors Th_1_ and Th_2_ in the modified bridge with IC_1_ OP amplifier for the linearization of output temperature T_2_ (temperature to voltage convertor).

**Figure 10 sensors-24-03547-f010:**
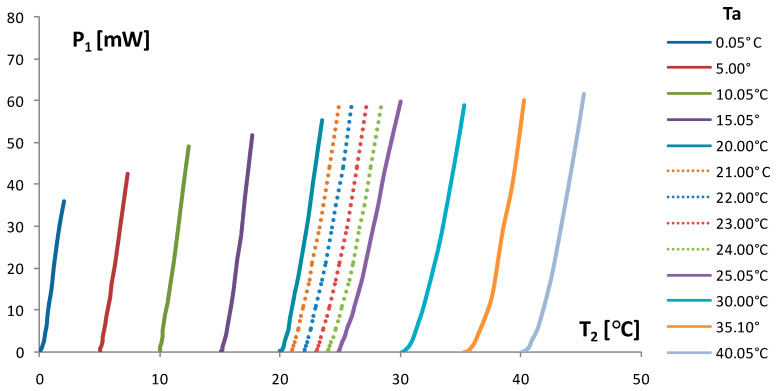
Input power P_1_ (DC regime) of self-heating thermistor Th_1_ as a function of output temperature T_2_ of the thermistor Th_2_. T_a_—ambient temperature as a parameter (solid curves). The dashed curves are interpolated using a polynomial of the third order. The limitations of self-heating current I_1_ are based on the criterion of stability k > 0.97 in saturation regime.

**Table 1 sensors-24-03547-t001:** Comparison of TF TCT, DISK TCT, and TCCT device properties.

Properties	TF TCT	DISK TCT	TCCT
R_0_ [Ω] nominal resistance at 20 °C	562	586	2600
B [K] temperature exponential factor	3353	3186	3908
type of device geometry	planar	cubic	cubic
device dimensions [mm]	25.4 × 12.7 × 1	D = 4.6; l = 6.5	2.2 × 1.4 × 1
V [mm^3^] device volume	323	108	3.08
S [mm^2^] occupied surface on board	323	30	3.6
T_a_ [°C] ambient temperature	1 to 40	1 to 40	1 to 40
U_max_ [V] supply voltage	16 to 8	14 to 4.5	14 to 5
I_1max_ [mA] self-heating current	15 to 48	16 to 24	2.5 to 9.5
P_1max_ [mW] nominal input power	500	450	100
R_1min_ [Ω] input resistance	905 to 155	956 to 105	5378 to 461
R_1max_ [Ω] input resistance	1375 to 173	1412 to 384	6399 to 1091
R_2min_ [Ω] output resistance	9855 to 1550	11,450 to 3060	5843 to 743
R_2max_ [Ω] output resistance	13,950 to 2992	11,950 to 3750	6499 to 1034
ΔT[K] temperature difference	6 to 10	4.5 to 18	2 to 16
t_d_ [s] self-heating time *	10 to 30	10 to 30	** 10 to 30
k_1,2_ relative thermal coupling	0.55	0.35	0.25
DC Insulation resistance [Ω] *	>10^12^	>10^9^	>10^9^
insulation resistance [Ω] * at 50 Hz	>200 M	>250 M	>200 M
parasitic capacitance [pF] *	<12.5	<3	<1

* measured at room temperature (20 °C), ** measured at 20 °C, U and I_1_ given in bar diagram in Figure 7.

**Table 2 sensors-24-03547-t002:** Comparison of TF TCT, DISK TCDT, and TCCT device sensitivity.

Properties	TF TCT	DISK TCT	TCCT
T_a_ [°C] ambient temperature	1 to 40	1 to 40	1 to 40
Δ_1min_ [K/V] min sensitivity	0.116 to 0.321	0.112 to 0.308	0.076 to 0.362
Δ_1max_ [K/V] max sensitivity	0.927 to 2.573	0.625 to 1.562	0.201 to 1.45
Δ_2min_ [mW/°C] power sensitivity	15.9 to 12.25	37.15 to 18.51	1.011 to 1.31
Δ_2max_ [mW/°C] power sensitivity	39.2 to 29.35	60.9 to 41.66	4.95 to 7.85

The inaccuracy in measurements of chip TCCT device response such as output temperature T_2_ depends on partial inaccuracies such as the following: Δ_T_ of temperature T, Δ_MS_ of supply voltage U, self-heating current I_1_, and resistance R_2_, Δ_B_ of exponential factor B and Δ_SH_ of using the Steinhart–Hart equation in the first approximation.

## Data Availability

Data are contained within the article.

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
