# Peer review of "Thermally Coupled NTC Chip Thermistors: Their Properties and Applications"

_sensors, 2024, doi:10.3390/s24113547_

Round 1
Reviewer 1 Report
Comments and Suggestions for Authors
General comment: The paper is a first-class "down-to-earth" engineering work with high value for all practitioners in this field and related applications. None-the-less, some little improvements are suggested most of which are formal:
- Weakness of expression: "the conductive electrical noise "? Can an electrical noise be conductive (below Fig. 1)? The wish of the authors might be to express that the quality of a direct temperature measurement could be affected by the high noise level as produced by power switching circuits. However, by making use of two independent properties of the coupled NTC system, namely (1) the comparatively low (electrical) coupling capacity between the two NTCs and (2) their thermal inertia, it allows (1) to reduce the disturbing influence of voltage or current peaks on the temperature measurement and (2) to smoothen the ripple of the temperature signal. Both effects could be meaningful for supervising the component's temperature or for other controlling purposes.
- "... nominal resistance around 2600 Ω ..." Here "at room temperature" (or similar) should be added.
- In Fig. 2 the notation >R*< is used but not explained in the figure caption (little printing error).
- Equation (2) does not contribute to understanding. Equation (3) and (4) can be derived directly from Equation (1) just by using Equation (1) twice; one time with T=T1 (and therewith R(T) = R(T1) = R1) and a second time with T=T2, respectively.
However, important is seemingly that B was not used from the data sheet (as could be assumed by the reader, since authors anticipate the B value already in the beginning of their discussion) but instead they performed two calibration measurements yielding two value couples (T10/R10 and T30/R30) as result which were used to derive the (by themselves experimentally established) B value. But this should be expressed verbally, if true, instead of rewriting equations which are mathematically redundant as they all express the same physical thing. More directly would be to list the measured values of R10 and R30.
- Temperature differences (as e. g. in the discussion of accuracy but also for temperature steps) should be expressed in K (not °C; in the entire paper).
- "А capacitor such as the 100nF on placed in parallel ..." Suggestion for improvement: "А parallel connected capacitor such as the 100nF ..."
- "Ta - ambient temperature as a parameter." is mentioned twice in the caption of Fig. 10 (delete one).
- Line 251, Typo: "dependant" -> dependent
- Lines up to 304: It does not come clear what the purpose or practical meaning of the derived dissipation length da is. What is the gain of knowledge in knowing a hypothetical heat dissipation length? At this point it should not be concealed that this parameter has been derived by very raw simplifications like neglecting that the surrounding air does not stand still but will move by natural convection (which can make a significant difference to still air). Of course, such a simplification like replacing the complicated heat transfer through natural convection by the physically much simpler heat diffusion might be justified. But in the current discussion the purpose of this dissipation length and also this justification is missing which - in addition - would need to take into account validity boundaries. (One purpose could be that any other heat source within the vicinity of the TCCT (the "size" of this vicinity is defined by da) could affect the characteristic behavior of the TCCT and therefore, this da value must at least be roughly estimated).
Line 328/329: An engineer would define a sensitivity as a relation of the effect and its cause. Surprisingly, here it is the other way around, since T2 (and so delta T2) is the reaction (effect) on P1 (delta P1, cause). Why? Maybe this could be shortly highlighted.
See above comments.
Author Response
Thank you for the comments, suggestions and requirements to improve the manuscript. Corrections are highlighted with green color. Please see the attachment.

Reviewer 2 Report
Comments and Suggestions for Authors
1. I cannot understand what do you mean "0,020°C of inaccuracy" and the number below Ta (for example 0,05°C) for Figures 5-7.
2. I think your NTC chips are obtained from a company, the properties of the NTC chips can be obtained from the company. I do not think the measurement of Figure 2 and the fitting equations in page 3 are OK?
3. The fitting curve can be obtained by using some figure drawing software, for example, "original".
4. The measured frequency of the results in Figure 10.
5. The heat dissipation of the circuit board should be taken into consideration.
6. What is your IC1 number?
Author Response
Thank you for your comments and suggestions to improve the manuscript. Please see the attachment.

Reviewer 3 Report
Comments and Suggestions for Authors
The article is acceptable after modification.
Comments on the Quality of English Language1. In Figure. 7, under the test conditions, when the voltage U increases, the time required for the self-heating current to stabilize gradually becomes longer. When the maximum test result of the voltage U is 8V, the time required for the self-heating current to stabilize is about 30s. As mentioned in Table 2, TCCT devices support a maximum voltage of 14 to 5V and a self-heating time of 10 to 30s. When the voltage is 8V to 14V, how is the self-heating current stabilization time calculated?
2. It can be found in Figure 8 that the resistance value of R2 is constantly changing with the change of time, and is not in a stable value. When the output resistance is unstable, how can it be used as a potentiometer in the application mentioned in the article?
3. As shown in Figure 4, the changes of self-heating current and power supply voltage at different ambient temperatures, the set power supply voltage increases by 1V every 20s, and it is indicated in the following content that it takes 10 to 30s for the change of self-heating current to reach stability. Then when the self-heating current does not reach stability at 20s, is the change of self-heating current measured at this time accurate?
4. Reference 6 and reference 20 are the same.
5. Some parameters, such as (Re), are not defined.
6. There are some grammatical and formatting problems (such as Spaces, and in any combination of upper and lower case) in the article. The authors should improve the English language thoroughly.
Author Response
Thank you for your comments and suggestions to explain TCCT stability and improve the manuscript. Please see the attachment.

Round 2
Reviewer 2 Report
Comments and Suggestions for Authors
Accept as it is.